# A Miniaturized Electrostatic Precipitator Respirator Effectively Removes Ambient SARS-CoV-2 Bioaerosols

**DOI:** 10.3390/v14040765

**Published:** 2022-04-06

**Authors:** Rachel K. Redmann, Brandon J. Beddingfield, Skye Spencer, Nicole R. Chirichella, Julian L. Henley, Wes Hager, Chad J. Roy

**Affiliations:** 1Infectious Disease Aerobiology, Division of Microbiology, Tulane National Primate Research Center, Covington, LA 70433, USA; rredmann@tulane.edu (R.K.R.); bbedding@tulane.edu (B.J.B.); skyemacp@gmail.com (S.S.); nchirichella@tulane.edu (N.R.C.); 2Section of Otolaryngology, Yale-New Haven Hospital, New Haven, CT 06520, USA; julianh@henleyion.com; 3Henley Ion, New Orleans, LA 70115, USA; 4Phase Three Product Development, Fort Collins, CO 80526, USA; whager@phasethreedev.com; 5Department of Microbiology and Immunology, Tulane School of Medicine, New Orleans, LA 70112, USA

**Keywords:** SARS-CoV-2, bioaerosols, electrostatic precipitation, respiratory protection

## Abstract

The inhalation of ambient SARS-CoV-2-containing bioaerosols leads to infection and pandemic airborne transmission in susceptible populations. Filter-based respirators effectively reduce exposure but complicate normal respiration through breathing zone pressure differentials; therefore, they are impractical for long-term use. Objectives: We tested the comparative effectiveness of a prototyped miniaturized electrostatic precipitator (mEP) on a filter-based respirator (N95) via the removal of viral bioaerosols from a simulated, inspired air stream. *Methods*: Each respirator was tested within a 16 L environmental chamber housed within a Class III biological safety cabinet within biosafety level 3 containment. SARS-CoV-2-containing bioaerosols were generated in the chamber, drawn by a vacuum through each respirator, and physical particle removal and viral genomic RNA were measured distal to the breathing zone of each device. Measurements and Main Results: The mEP respirator removed particles (96.5 ± 0.4%), approximating efficiencies of the N95 (96.9 ± 0.6%). The mEP respirator similarly decreased SARS-CoV-2 viral RNA (99.792%) when compared to N95 removal (99.942%), as a function of particle removal from the airstream distal to the breathing zone of each respirator. Conclusions: The mEP respirator approximated the performance of a filter-based N95 respirator for particle removal and viral RNA as a constituent of the SARS-CoV-2 bioaerosols generated for this evaluation. In practice, the mEP respirator could provide equivalent protection from ambient infectious bioaerosols as the N95 respirator without undue pressure drop to the wearer, thereby facilitating its long-term use in an unobstructed breathing configuration.

## 1. Introduction

SARS-CoV-2 is a pathogenic beta-coronavirus that is the source of a current worldwide pandemic [1]. The virus has thus far proved to be highly transmissible and passes easily via respiratory droplets between shedding from infectious to naïve hosts [2]. The combination of the inability to readily kill immunocompetent hosts and instead use the respiratory system as an efficient vectoring pathway to infect others, has made the resultant disease (COVID-19) one of the most prolific in human history, infecting hundreds of millions, and killing >5 million worldwide [3]. There has been a sustained surge in the use of respiratory protection throughout all sectors of society to effectively reduce ambient exposure and avoid infection [4,5,6]. The majority of the public currently use facial coverings consisting of cloth or ‘surgical’ masks, which provide protection only to others proximal to the wearer of the cloth mask [7,8,9]. The mechanism of particle collection for cloth masking is through the microburst high-velocity exhalation of the wearer and subsequent impaction upon inside surface of the mask [10]. Although essentially no protection from ambient bioaerosols is provided when donning cloth/surgical masks, current public health mandates during the pandemic require the use of cloth/surgical masking by the public to further reduce community disease burden by retarding respiratory transmission at the source generator (the infected wearer). Some among the public and most healthcare professionals have chosen to don filter-based respirators which, unlike cloth masks, provide respiratory protection from ambient bioaerosols through impaction and interception collection on a thermospun/cellulose filter substrate [11]. The most popular of filter-based respirators is the nonoil-95 percent collection (N95) filter-based respirator. For filter respirators to properly function, inspired air is required to pass through the face of the filter respirator and a tortuous matrix of the substrate before reaching the breathing zone of the wearer [12,13]. Particle collection by the filter respirator is also contingent upon a nearly airtight seal on the face of the wearer [14,15]. This configuration in properly donned filter respirators produces a significant pressure drop (~0.27 ∆P cm H_2_0) between ambient air and the breathing zone upon inspiration. While providing sufficient protection against particles of a particular size (>0.3 μm non-oily aerosols), this pressure drop can become noticeable and laborious during longer-term use in otherwise healthy users and may become obstructive in individuals with compromised respiratory systems.

Particle collection by electrostatic precipitation (EP) has been used successful in industrial applications for decades [16], although historically, the application of this technology for the purposes of personal respiratory protection has not been pursued, with a few exceptions of harnessing a form of EP on a miniaturized scale for personal protection [17,18,19,20]. The principle of EP allows for the removal of particles from an airstream based upon the coronal discharge of electrons onto an opposite-charged collection plate. The efficiency by which this mechanism collects airborne particles is correlative to the amount of energy provided to the corona, transit time, distance, and corresponding flow rate of the targeted airstream. The most commercially available ionizer-based devices use a corona discharge emitter creating an excess of ions and electrons interposed into the airflow followed by collector plates separated by a specific distance and electric field density between them. The airborne incidental particles and aerosols acquire a charge and are then deposited upon the collector plate and diverted from their intended path. Most of those designs have a unidirectional functionality [17] and often have supportive fans to maintain the desired air flow past the corona discharge and between the collector plates needed to remove the particles from the air flow path. In this application, an EP was miniaturized (mEP) and engineered to collect particles from the uninterrupted respiratory airstreams of the wearer. This design also relies upon natural velocities of normal inspiratory and expiratory flow to generate the airflows required to bring functionality to the device. The resulting mEP respirator functions as an energized particle collector without any appreciable pressure drop between ambient air and the breathing zone of the wearer because the mechanism of particle collection is not filtration, a mechanism that requires wearer inspiratory flow to actively pull and push air through a filter substrate. A prototype of the mEP respirator was engineered to be self-contained and worn in a similar configuration that approximates the face fit and head strapping design of popular filter-based respirators.

In this study, we tested the particle collection efficiencies of the prototyped mEP respirator and compared performance with a N95 filter respirator, and an in-line HEPA cartridge filter. Each of the devices were tested using a modified 16 L chamber operated in a dynamic configuration within a high (BSL-3) biocontainment [21]. An atmosphere of viral bioaerosols was synthetically generated and maintained in the chamber during the testing of each device. After fully characterizing the particle concentration, airborne viral titer, and corresponding counts within the ambient environment of the chamber, we measured the collection efficiency of each device. Once configured into the chamber, each respirator was sealed in such a way to only collect airstream contents distal to the inlet face or ‘breathing zone’ upon inspiration. Total particle counts were performed initially to determine the particle reduction for each device. Thereafter, aerosol samples were collected from the airstream in the same configuration and analyzed for culturable virus by TCID_50_ and genomic RNA content as a surrogate for viral capture, a constituent of the bioaerosols collected by each respirator.

## 2. Materials and methods

### 2.1. Design and Operational Principles of the mEP Respirator

The prototype device (Figure 1a) is designed to fit on the face of the user and is held to the face by strapping to the back of the head. The prototype device was designed to minimize corona discharge by setting an operational point and controlling with an embedded CPU control of the emitter voltage (Figure 1b,c) so that effective particle reduction can occur within the specified geometry, and very little corona discharge or ozone formation is encountered. This servomechanism maintains a performance at a set point despite ambient temperature, humidity, or site elevation. Therefore, it is advantageous to maintain control of the ionization process and assure its performance under different circumstances and elevations. The same servomechanisms allow for the fine control of ozone (O_3_) production with or without a catalytic degradation filter.

### 2.2. Benchmark Testing of the mEP for Particle Removal

Preliminary testing of the particle removal capacity of the mEP was performed using aerosolized salt solution at a high inlet flow (85 L per minute (LPM)). Briefly, an aerosol particle generator (Model 8026 Particle Generator, TSI Systems, St. Paul, MN, USA) loaded with saline solution was connected to a test chamber via tubing and passed through honeycomb to facilitate laminar flow through the chamber that housed the device and the samplers. A regenerative blower (VFC 080P-5T, Fuji Electric, Edison, NJ, USA) was used to create airflow (85 LPM; 16” H_2_0) through the mEP device while housed in the chamber. Flow through the mEP was monitored during this assessment (SFM3000 Mass Flow Meter, Sensirion AG, Stäfa, Switzerland). Particle counts were measured distal to the inlet of the mEP using an optical particle counter (Model 8525, TSI). Ozone concentration was measured using an ozone measurement device (FD-90A-O3 Forensics Detectors, Estates, CA, USA). Power requirements to the mEP device during this preliminary assessment were both supplied to the device and measured (using a multimeter) in relation to the active particle removal and ozone measurements. The results showed (Figure 1d) that a ~0.35 milliwatts (mw) power selection resulted in a particle collection of ~95%. The results of time series experiments, using longer operational durations, indicated that particle rejection slightly improved to ~97.5%, suggesting relatively stable particle collection over extended wear times (data not shown). Similarly, O_3_ levels generated by the device (Figure 1e) were below U.S. Occupational Safety and Health Administration (OSHA) limits of <0.1 parts per million (ppm), and at the selected power level of 0.35 mw, O_3_ levels remained below the U.S. Food and Drug Administration (FDA) limits of <0.05 ppm. The results of the saline aerosol particle removal testing ensured that the operational functionality and subsequent evaluation involving SARS-CoV-2-containing bioaerosols were performed within biocontainment. Performing benchmark testing using harmless saline aerosols also allowed for the measurement of O_3_ (which was not measured in biocontainment), and optimized the mEP power setting for the removal of viral bioaerosols when the device was used in containment.

### 2.3. Experimental Airborne Viral Removal Efficiency

#### 2.3.1. Experimental Configuration

Use of standard mask evaluation methodology (e.g., 42 CFR Part 84) [22] was not feasible for the experimental approach as SARS-CoV-2 aerosolization required additional engineering controls, including housing our configuration within a Class III biological safety cabinet for added safety during these studies because of the biologically active nature of the virus and the necessity of biosafety level 3 (BSL-3) containment. Therefore, all SARS-CoV-2 aerosol generation took place inside a 16 L polycarbonate chamber outfitted with dilution and exhaust tubing and a sampling orifice. The chamber was connected to an automated system (Biaera Technologies, Hagerstown, MD, USA), which controlled dilution, exhaust, sampling, and generator air flows when applicable, and recorded temperature, relative humidity, and pressure readings. The automated system maintained equal rates of total air flow in and out of the chamber to retain equilibrium. Figure 2 illustrates the experimental configuration of the chamber utilizing the mEP respirator and the various sampling strategies implemented within biocontainment. The aerosol generator (Figure 2A) used was the 3-jet (3JC) collision nebulizer. Samples from the chamber (Figure 2C) were collected using either the APS for particle counting (Figure 2D) or the all-glass impinger (Figure 2E) sampler for virus collection. Total air flow in and out of the system was maintained at 16 LPM, with adjustments to the dilution (Figure 2B) and exhaust flows (Figure 2F), as needed for differing generator and sampling requirements, were facilitated through the use of an automated aerosol control system (Biaera).

#### 2.3.2. Experimental Procedure

Individual aliquots of a liquid volume (5 mL) of SARS-CoV-2 inoculum and corresponding dilutions were prepared for evaluation of each respirator. Upon performance of each respirator, a liquid aliquot was directly expressed into the precious fluid reservoir by the Collison nebulizer, then actuated and allowed to continuously run for analysis. The experimental configuration was harmonized amongst both respirators and shared a similar design and internal volume (Figure 1). Flow rates for the configured system were maintained at 16 LPM. The stability of the system was dictated by the overall flow from the nebulizer, and dilution air added to the flow for a total of 16 lpm was operated in a dynamic fashion in a 16 L chamber under constant exhaust, which was balanced by differential pressure transduction and the automated control of mass flow controllers for both the input and outputs of the system. Thus, the stability of the ambient concentration of the particles was expected to reach a steady state in approximately 1 min (1 air change/min in chamber). Two aerosol sampling instruments with differing flows were used in each discrete aerosol generation event. For the experiments involving particle counting, the aerodynamic particle sizer (Model 3321, TSI) was used, which housed an internal exhaust flow of 5 LPM. Particle counting within the system respirator exposure chamber yielded a relatively stable generation upon initiation and throughout testing (Appendix A). Residual exhaust flow was provided via an external pump at 2 LPM. SARS-CoV-2 aerosols were also collected in separate aerosol generation events for the purposes of the biological viability determination of the viral aerosols. Individual all-glass impingers (AGI-4, SKC, eighty-four, PA) were used to collect aerosol samples from either the ambient chamber or the flow distal to each respirator and were actuated upon initiation of each run of the aerosol system. The AGI-4 sampler requires 6 LPM exhaust flow for operation. The residual exhaust flow from the chamber was adjusted according to either sampling requirements or the necessity to maintain neutral pressure (0” H_2_0), which was actively monitored throughout all aerosol generation events. The dynamic flows, as described through the evaluation chamber, were continuously operated for every evaluation of each respirator. Temperature and humidity were continuously monitored. The prevailing temperature was 20.4 ± 3.6 °C, and the relative humidity was 57.6 ± 7.2% across all evaluations.

#### 2.3.3. Measurement with Aerodynamic Particle Sizer

Particle characteristics, including particle counts, were determined using an aerodynamic particle sizer (APS Model 3321, TSI Inc., St. Paul, MN, USA). The APS measured the aerodynamic size of particles from 0.5–20 microns and used time-of-flight analysis based upon the velocity and relative density of interrogated particle stream to determine the behavior of airborne particles. Aerosol was drawn into the APS at a total flow of 5 LPM; 20% of the total flow was dedicated to the inlet into the analyzer; 80% was sheath flow. The APS spectrometer used a double-crest dual laser system and nozzle configuration, which reduced the advent of false (e.g., doublet) background counts. Particle resolution by this instrument was thoroughly characterized, demonstrating <2% in geometric standard deviation when sampling monodisperse aerosol distributions [23]. Resolution time of this instrument was 1 scan/s. Analysis of data from the APS was conducted, and a device software (Aerosol Instrument Manager Version 5.3, TSI Inc., St Paul, MN, USA) was used for initial review of data. The APS device operated on a continual basis once aerosol generation was initiated and logged data for the duration of each aerosol event. The resulting particle size generated by the Collison nebulizer in this configuration yielded a mass median aerodynamic diameter of 1.3 µm and geometric standard deviation of 1.4 (Appendix A).

#### 2.3.4. Virus and Cells

Virus used for aerosol generation was strain SARS-CoV-2; 2019-nCoV/USA-WA1/2020 (BEI# NR-52281) prepared on subconfluent VeroE6 cells (ATCC# CRL-1586) and confirmed via sequencing. Vero E6 cells were used for live virus titration of sample input and were maintained in DMEM (#11965092, Thermo Scientific, Waltham, MA, USA) with 10% FBS.

#### 2.3.5. Quantification of Viral RNA in Aerosol Samples

Viral RNA in collected aerosol samples was quantified using RT-qPCR targeting the nucleocapsid (genomic) of SARS- CoV-2. RNA was isolated from aerosol samples using a Zymo Quick RNA Viral Kit (#R1035, Zymo, Irvine, CA, USA), per manufacturer′s instructions. RNA was eluted in RNAse-free water and was extracted using 100 μL of sample. Isolated RNA was analyzed in a QuantStudio 6 (Thermo Scientific, USA) using TaqPath master mix (Thermo Scientific, USA) and appropriate primers/probes [24] with the following program: 25 °C for 2 min, 50 °C for 15 min, and 95 °C for 2 min followed by 40 cycles of 95 °C for 3 s and 60 °C for 30 s. Signals were compared to a standard curve generated using in vitro transcribed RNA of each sequence diluted from 10^8^ down to 10 copies. Positive controls consisted of SARS-CoV-2-infected VeroE6 cell lysate. Viral copies per sample were calculated by multiplying mean copies per well by amount in the total sample extract.

#### 2.3.6. Quantification of Culturable Virus

Median tissue culture infectious dose (TCID_50_) was used to quantify replication-competent virus in viral stock used to generate the aerosols for respirator evaluations. VeroE6 ells were plated in 48-well tissue-culture-treated plates to be subconfluent at time of assay. Cells were washed with serum-free DMEM, and virus from 50 μL of sample was allowed to adsorb onto the cells for 1 h at 37 °C and 5% CO_2_. After adsorption, cells were overlayed with DMEM containing 2% FBS and 1% Anti/Anti (#15240062, Thermo Scientific, USA). Plates were incubated for 7–10 days before being observed for cytopathic effect (CPE). Any CPE observed relative to control wells was considered positive and used to calculate TCID_50_ by the Reed and Muench method.

#### 2.3.7. Statistics

All data from the evaluation studies that included particle counting and viral bioaerosol removal were assessed using Prism 9 (GraphPad Software, San Diego, CA, USA). For both sets of data, a repeated measures ANOVA was performed with Geisser–Greenhouse correction and Tukey′s multiple comparisons test. Significance was noted at *p* < 0.05.

## 3. Results

The results of the particle counting in the experiments involving viral bioaerosols showed a clear reduction in the number of particles reaching the distal portion of the mask when the powered mEP or N95 respirator was in place (Figure 3). A HEPA filter was used as a control mechanism to confirm particle counts, which were essentially zero when positioned to receive inspiratory flow. The mEP was performed on approximately the equivalent basis as the N95 respirator, returning an average 96.5% particle removal compared to the N95, which removed an average of 96.9% of particles at measured air flow velocities. There was no significant difference between the particle removal rates between the mEP and the N95 respirator. There were significant differences when the ambient particle concentration was compared to the powered mEP, N95, or HEPA filter values (Figure 3).

The results of the viral bioaerosol evaluation indicated, as observed from the particle removal experiments, that the mEP removed most or all of the viral RNA from the airstream, as measured by the RT-PCR analysis of the distally positioned aerosol sampler (Figure 4). When the mEP was de-energized (denoted in the legend as mEP ‘OFF’), viral RNA approximated ambient levels were expressed in SARS-CoV-2 genome copies/liter aerosol. Interestingly, percentage removal, when measured by viral RNA, was remarkably similar between the mEP (99.79%), N95 (99.94%), and the HEPA filter (99.99%).

## 4. Discussion

SARS-CoV-2 is an emergent, highly transmissible coronavirus. It is the etiologic agent for COVID-19, and the source of an ongoing worldwide pandemic. Over 5.4 million people have died worldwide from COVID-19 to date, with more than 847,000 deaths in the United States alone. Infection is associated with highly heterogenous disease sequelae, ranging from completely asymptomatic to severe acute respiratory distress and death. During this pandemic, respiratory protection in the form of facial coverings and, at times, filter-based respirators has been promoted to curb viral transmission and mitigate disease impact. Here, we performed a preliminary evaluation of a respirator that utilizes miniaturized electrostatic precipitation (mEP) rather than filtration for aerosol particle removal from the breathing zone of the wearer. We demonstrate that the mEP-based respirator in this evaluation effectively removed laboratory generated SARS-CoV-2-laden aerosol particles from an airstream at a rate that approximates filter-based respirators. The mEP respirator achieved the equivalent particle reduction without the pressure decreased required of filter-based respirators, such as the N95. These data underscore the prospect for the use of the new technologies that rely upon an alternative mechanism of airborne particle removal from the airstream, rather than substrate-based filtration.

### 4.1. Significance

In this study, we used an mEP respirator specifically designed to remove aerosol particles from the airstream at a rate associated with normal human respiration. The respirator was engineered to remove airstream particles during both inhalation and exhalation, although studies in this evaluation only sampled and measured inspiratory flow. The mEP respirator removed aerosol particles as demonstrated by the reduction in particle counts, and the particle counter was configured distal to the ambient air inlet and functional mEP within the respirator. The measurement of viral removal as a correlative of physical particle removal was measured in the liquid impingement samples also collected distal to the respirator inlet and mEP device. The viral concentration of the ambient air surrounding the respirator was also sampled by liquid impingement for the purposes of residual comparison. The viral content of these liquid samples was measured by median tissue culture infectious dose (TCID_50_) and quantitative RT-PCR. The resulting culturable concentration of SARS-CoV-2 in the ambient air of the exposure chamber averaged 3.4E + 3 TCID_50_/L, or 9.7E + 5 genome copies/L of aerosol. The TCID_50_ measurements of all liquid impinger samples distal to the mEP respirator inlet and the N95 respirator were nondetectable. This was partly due to the poor sensitivity of the TCID_50_ assay and the use of the cell culture to attempt to quantify low titer virus. Fortunately, all impinger samples were split after collection and processed for the analysis of the genomic content by RT-PCR. The results of this sampling showed an approximately equivalent removal of genomic material by the mEP and the N95 respirator under a similar laboratory configuration. The reduction in SARS-CoV-2 genomic content by either method was highly correlative with the physical particle removal resulting from particle counter measurements. Liberated SARS-CoV-2 virions (≈110 nm) are componentry of biological aerosol particles and will not travel in the air alone but as a component of a larger particle. In this case, aerosol particles generated in this evaluation were ≈2 μm, thus any SARS-CoV-2 virions could inhabit the particles removed by the mEP.

### 4.2. Limitations

There are several limitations in the preliminary evaluation study performed on the mEP respirator since the generation of SARS-CoV-2 bioaerosols necessitated the use high-containment (biosafety level 3) laboratory environment and unique aerobiology facilities configured for studies with high-consequence pathogens and not necessarily respirator efficacy testing. The configuration conventionally used to test efficacy (NIOSH standard 42 CFR 84, also referred to as “Part 84”) requires the use of a high-flow input (≈85 LPM) as inlet flow across the face of the filter substrate is tested for removal efficiency. The high flow simulates a pressure drop associated with the velocity of inspiration experienced by the user when breathing through a filter-based respirator. The mEP respirator is configured, whereas there is a nearly imperceptible pressure decreases upon inspiration; therefore, there is no utility for the use of high flows for the purposes of evaluation. Rather, the inlet flows used in this evaluation approached, but did not exceed, 6 LPM during the studies. In addition, the biocontainment laboratories where the infectious aerosols were used discouraged use of high flow rates (>20 LPM) within the system in use. Therefore, the evaluations were limited to a sampling rate that approximated the lower threshold of respiratory minute volume in all the respirators evaluated. The disparity between the flowrates used in the NIOSH testing and this evaluation can be considered a weakness of the study and contrasts with the higher flow rates used in previous studies on masks [25]. Similarly, although the mEP and the N95 respirators were tested under the inspiration configuration (6 LPM, either across the filter substrate of the N95, or through the inlet of the mEP), neither respirator was tested for its efficacy for particle removal efficacy upon expiration. Although the mEP respirator theoretically removes particles upon expiration, this evaluation study does not include a demonstration of the technology performing in this manner, and this should be one of the aspects of future evaluation studies.

Viral bioaerosols removed from the inspiratory airstream accumulate on the collector plates of the mEP respirator within the device. There was no analysis to determine the viability of the virus collected on the mEP plates. The particle removal efficacy studies did not include a hygiene protocol to elute collected aerosol particles from the collector plates once removal had taken place. Theoretically, a virus captured in this manner (electrostatically) would dehydrate and be rendered inactive, and although this study did not include a laboratory demonstration of this effect, previous studies using electrically charged surfaces illustrate the capacity for the deactivation of a virus [26]. Similarly, the mEP respirator required a continuous power source for functionality and performance, and although airflow was not restricted when the mEP unit was not powered, the benefits of particle removal dissipated. When powered, the mEP unit within the respirator generates ozone (O_3_) as a by-product of the coronal discharge of electrons and oxidation of ambient oxygen, with the amount of O_3_ generated directly proportional to the mEP power requirements needed to remove particles from the airstream at a desired percentage efficiency. Ambient O_3_ at certain concentrations is theoretically microbicidal, although this effect was not tested as an aspect of this evaluation. O_3_ can also be detrimental to human health when inhaled at relatively high concentrations, and the FDA requires the O_3_ output of indoor medical devices to be no more than 0.05 ppm or 50 μg/L air. The National Institute of Occupational Safety and Health (NIOSH) recommends an upper limit of 0.10 ppm (100 μg/L air), not to be exceeded at any time during an 8 h workday. The mEP in this respirator was engineered in a miniaturized format that provides a high percentage of particle removal at a relatively low-power requirement. The preliminary assessment of the mEP using a variety of power requirements showed a low O_3_ production at the power setting that was used in the viral bioaerosol experiments (0.35 mw). This design resulted in the generation of O_3_ under FDA limits and NIOSH recommended time-weighted average concentrations, although O_3_ measurement was not performed coincidental to the viral bioaerosol removal evaluation and should be included in future studies.

In summary, the results of this evaluation demonstrated an approximately equivalent performance in particle removal and viral RNA reduction for both respirators. The mEP respirator successfully demonstrated particle removal and viral removal as a constituent of the bioaerosol without a pressure decrease for the laboratory-simulated user. The mEP respirator, if further developed, could benefit individuals who require respiratory protection for long periods of time without the necessity of labored breathing through filter substrates when the use of other respiratory protection, such as a powered air-purifying respirator (PaPR) or self-contained breathing apparatus (SCBA), is not available or impractical. The mEP respirator could also be used in circumstances where respiratory protection is required for ambulatory or other compromised individuals who may not be physically able to produce the inspiratory flow required to breathe through filter-based respirators. Future evaluations should investigate many of the aspects that were identified limitations of this study, including the effect of O_3_ generation upon the efficiency of particle removal, as well as the laboratory assessment of the pathogen-agnostic mechanism of particle removal for other airborne threats such as influenza.

## Figures and Tables

**Figure 1 viruses-14-00765-f001:**
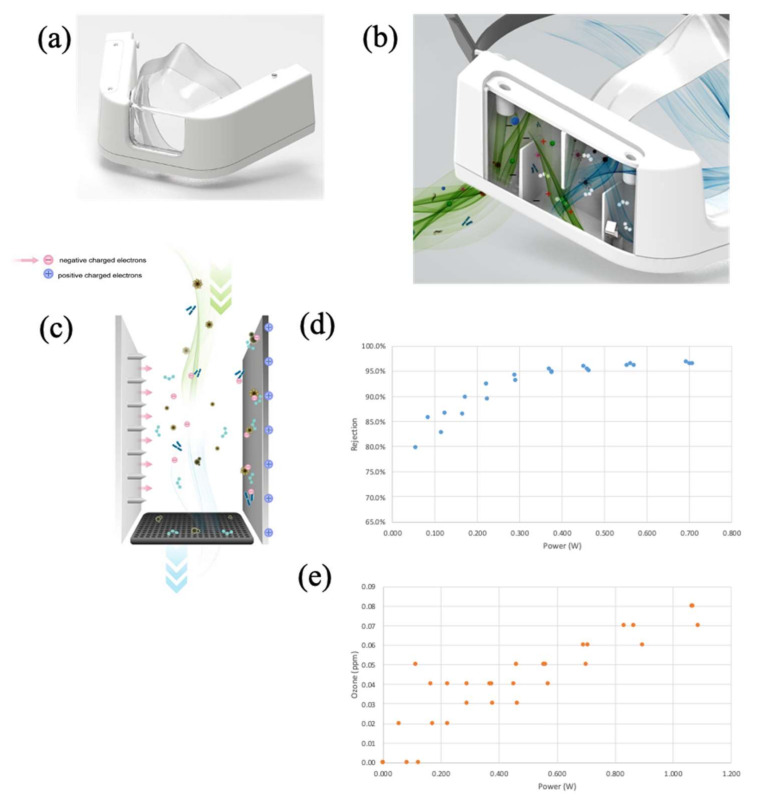
Protoyped mEP respirator. (**a**) Front view, picture. (**b**) Three-dimensional computer rendering with side panel cover omitted to reveal internal mechanics of particle removal from inspiratory flow: green-colored air flow represents ambient air from inlet into mask; blue-colored air flow denoted air with particles removed by the mEP respirator. (**c**) Three-dimensional computer rendering exploded view of the mEP unit within the mask, promotion of charged electrons and electrostatic precipitation, and collection plate. (**d**) Analytical determination of power and corresponding particle removal percentage when operated at 85 lpm flow through the respirator inlet. (**e**) Analytical determination of power requirements to endogenous ozone generation by the mEP.

**Figure 2 viruses-14-00765-f002:**
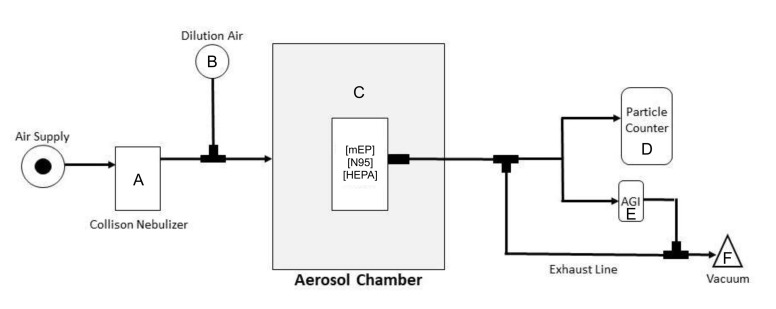
Configuration of exposure system used with SARS-CoV-2 bioaerosol evaluation. The air supply to the system was maintained at >40 PSIG and at ~21 PSIG to the nebulizer (**A**) which generated ~7 LPM at this pressure and dilution air (**B**) providing auxiliary air flow for mixing at 9 LPM. The total flow into the chamber was 16 LPM. The 16 L volume aerosol chamber (**C**) was dynamically operated with a constant flow of nebulized aerosol particles within the combination flow provided by the nebulizer and dilution air. Each device or filter was held in the center of the chamber by a modified ring stand, and the distal portion of each respirator sealed only the sampling port within the chamber in discrete experiments. An inlet for sampling by the particle counter (**D**) or AGI aerosol sampler (**E**) at an exhaust flow of either 5 or 6 LPM, respectively, was actuated at separate times, and residual exhaust, especially when the sampler was disengaged, provided a complete exhaust flow of 16 LPM (**F**). The entire system was expertly controlled using the AeroMP automated aerosol exposure system (Biaera).

**Figure 3 viruses-14-00765-f003:**
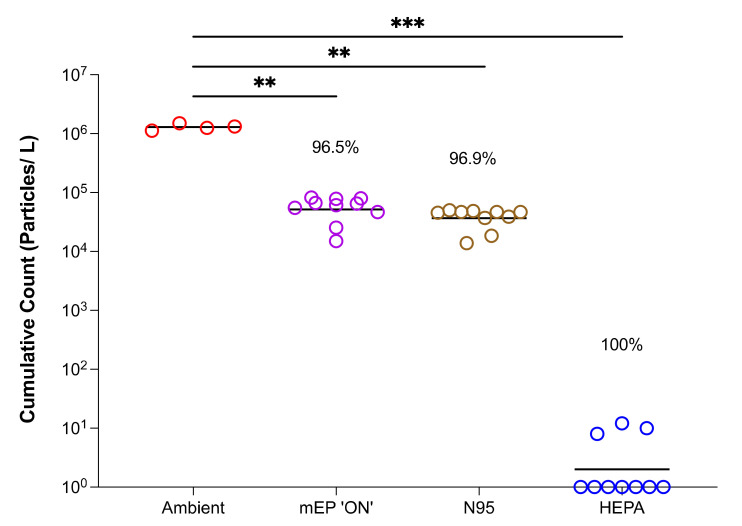
Particle counts of viral bioaerosols using the evaluation system within biocontainment. Counts from the energized mEP significantly reduced aerosol particles by a mean of 96.5% when compared to ambient (chamber) aerosol content; the N95 filter respirator significantly reduced particles by an approximately equivalent 96.9% when compared to ambient (chamber) particle content. The HEPA filter essentially removed all particles from the airstream. Colored open circles represent iterative run under named condition. Asterisks denote significance at *p* < 0.05 (**) or *p* < 0.005 (***).

**Figure 4 viruses-14-00765-f004:**
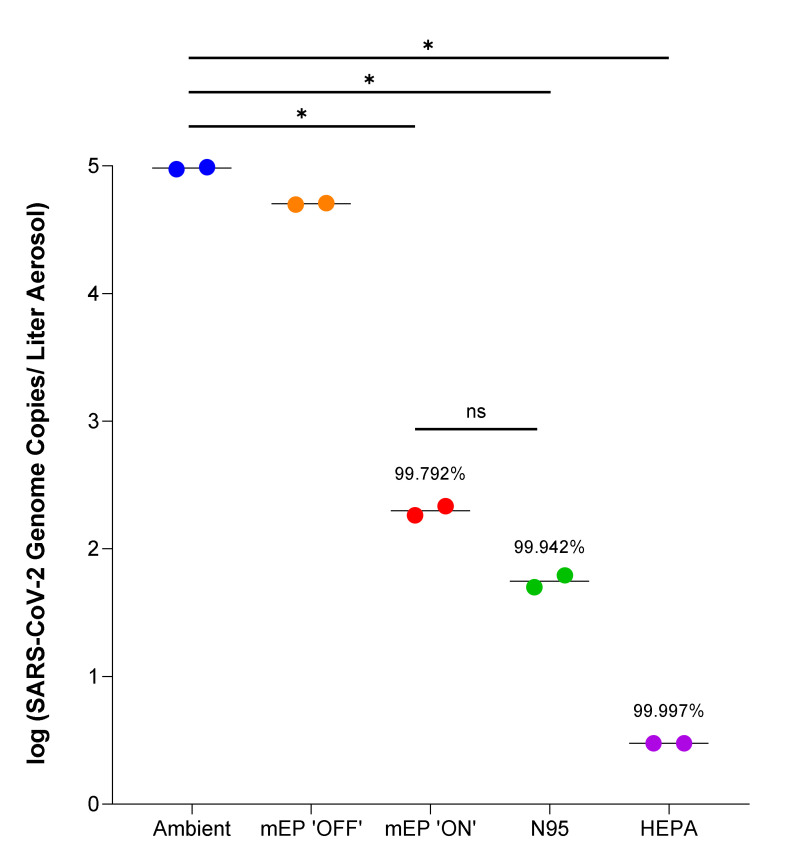
Removal of SARS-CoV-2 viral RNA using either the mEP or N95 filter respirator in the aerosol evaluation system within biocontainment. The energized mEP significantly removed an average of 99.792% SARS-CoV-2 viral RNA from the air when compared to ambient (chamber) viral RNA aerosol concentrations. The N95 significantly removed an average of 99.942% SARS-CoV-2 viral RNA when compared to ambient (chamber) viral RNA aerosol concentrations. The de-energized mEP removed an insignificant amount of SARS-CoV-2 viral RNA and the ambient (chamber) viral RNA aerosol concentrations. The in-line HEPA filter essentially removed all viral RNA from the airstream. Colored closed circles represent iterative run under named condition. Asterisks denote significance at *p* < 0.05 (*) or ns (nonsignificant).

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
