# Peer review of "A Miniaturized Electrostatic Precipitator Respirator Effectively Removes Ambient SARS-CoV-2 Bioaerosols"

_viruses, 2022, doi:10.3390/v14040765_

Round 1
Reviewer 1 Report
Please see the attachment.

Author Response
Reviewer 1
Major comments:
1. The mEP uses electrostatic precipitation to collect aerosols, please show the size specific particle collection efficiency under different breathing rate and pattern. The normal minute
ventilation is between 5-8 LPM and can exceed 20 and 40 LPM with light and heavy exercise, respectively. Thus, it would be nice to see if mEP can be tested at 20 LPM at least, which is also seen in other facemask studies (Kulmala et al. 2021).
The experimental configuration in Biosafety Level 3 laboratories discourages use of high flow systems when aerosolizing fully pathogenic virus, thus the flows for the purposes of
sampling (6 lpm) approximated minute volume associated with normal respiration. We have added detail about this limitation within biocontainment to our study design description (lines 336-339, under 4.2 ‘limitations’), and also added the suggested
reference (Kulmala et al., 2021) to our reference list as suggested testing conditions.
2. On page 5, line 184, please show the stability of the entire system (i.e., the stability of the generated particle size distribution and time series)?
We have added the following verbiage to description of the aerosol exposure system configuration in the context of overall stability. “The stability of the system was dictated
by the overall flow from the nebulizer, dilution air added to the flow for a total of 16 lpm operated in a dynamic fashion into a 16-liter chamber under constant exhaust that was balanced by differential pressure transduction and automated control of mass flow controllers on both the input and outputs of the system. Thus the stability of the ambient concentration of the particles expected to reach stable-state in approximately 1 minute (1 air change/min in chamber).” The details on the specific particle size is described at line 227-228.
Minor comments:
3. On page 3, Figure 1(c), the figure quality (e.g., I can’t see the negative charge electrons and positive charge electrons clearly) needs to be improved.
An improved figure with better resolution has been provided.
4. On page 5, Figure 2, please show more details how the N95 respirator were setup within the aerosol chamber. What is the flow rate of the N95? Please add a table comparing the flowrate,
pressure drop, size, weight and cost of mEP and N95.
We have added detail to figure 2 in the legend that describes flow rate from the distal sampling of the N95 and prototype mEP within the aerosol chamber. There is also
description of this system, including flow rates in lines 191-206 of the manuscript.
Pressure drop and other distinquishing characteristics such as size and weight were not
measured nor part of this study. Similarly, there were no estimates on the pricing of the mEP prototype provided to our laboratories as this was a research-only experimental
respirator at the time of testing.
5. On page 7, Figure 3, please explain what asterisk (*) means. Why are there fewer cumulative count for ambient condition?
The asterisks denote statistical significance between the two groups of data being compared as defined by the horizontal lines directly below the asterisk and by the statistical test described in lines 255-258. The statistically significant differences in particle counts shown in Figure 3 are described in 3.0 Results section, lines 268-271. We have added a reference to (Figure 3) in this description.
6. On page 8, Figure 4, please explain what asterisk (*) means.
The asterisks denote statistical significance between the two groups of data being compared as defined by the horizontal lines directly below the asterisk by the statistical test described in lines 255-258. The statistically significant differences in viral RNA shown in Figure 4 are described in 3.0 Results section, lines 272-274; Figure 4 is
referenced.
7. Please show the resolution time for the APS and optical particle counter.
Resolution estimates for the APS are available in the literature (Chen et al., 1985) and this reference has been added to the description of the equipment in the materials section.
Resolution for the optical particle counter used has been added to the manuscript from the users manual.

Reviewer 2 Report
I shall review this paper on its scientific merits and not from its practicality. The paper describes tests performed on a miniaturized electrostatic precipitator fitted into a mask to be worn in place of a N95 ask. Hence the tests are comparative between these two types of masks and a HEPA filter. The tests are performed under confined conditions using actual SARS Cov-2 viruses.
Standard aerosol particle sizing equipment is used for the tests.
The result is that the electrostatic mask is just about as efficient as the N95 mask, the HEPA filter being marginally better than both.
This is a very important paper and I recommend it for publication. Primarily as it shown that artificially applying electrostatic particle trapping (as opposed to natural electrostatic trapping due to triboelectrical effects in the N95 mask) allows the passage of virus particles to be stopped.
Ozone production by the corona discharge is discussed and the authors insist that the level is below health standards. This is perhaps disputable but from the pointy of view of the practical application of the technique. It is not a hindrance to the publication of this paper. Simply a factor that has to be kept in mind in subsequent designs.
While the experimental method is well explained, there is almost no information on the precipitator itself. (Voltages, method of ionization, dimensions, etc.). This makes the paper impossible to reproduce. This could be due to confidential industrial property concerns but it does lessen the usefulness of the paper. I feel however that this should not hinder its publication.
I am not sure about some of the terminology used and would recommend its modification. In particular the device is a “miniaturized” electrostatic precipitator and not a “micronized” device. (This refers to particle milling). Also “componentry” is used to refer the vehicles and machines, not virus particles. Another word should be sought. I have also noted that the efficiency for particle removal is slightly lower than the one for RNA as measured by RT-PCR, although very close. Is there any explanation for that?
Author Response
Reviewer #2
I shall review this paper on its scientific merits and not from its practicality. The paper describes tests performed on a miniaturized electrostatic precipitator fitted into a mask to be worn in place of a N95 ask. Hence the tests are comparative between these two types of masks and a HEPA filter. The tests are performed under confined conditions using actual SARS Cov-2 viruses.
Standard aerosol particle sizing equipment is used for the tests. The result is that the electrostatic
mask is just about as efficient as the N95 mask, the HEPA filter being marginally better than both. This is a very important paper and I recommend it for publication. Primarily as it shown
that artificially applying electrostatic particle trapping (as opposed to natural electrostatic trapping due to triboelectrical effects in the N95 mask) allows the passage of virus particles to be stopped. Ozone production by the corona discharge is discussed and the authors insist that the level is below health standards. This is perhaps disputable but from the pointy of view of the practical application of the technique. It is not a hindrance to the publication of this paper.
Simply a factor that has to be kept in mind in subsequent designs.
Authors thank the reviewer for the useful commentary on the nature of the paper and the prelimnary results. Considerations of these comments will be kept in mind in future studies.
While the experimental method is well explained, there is almost no information on the precipitator itself. (Voltages, method of ionization, dimensions, etc.). This makes the paper impossible to reproduce. This could be due to confidential industrial property concerns but it does lessen the usefulness of the paper. I feel however that this should not hinder its publication.
Authors concur with the reviewer, nearly all aspects of the engineering parameters and design are considered trade secrets to the developer and unfortunately do not appear in
this manscript.
I am not sure about some of the terminology used and would recommend its modification. In particular the device is a “miniaturized” electrostatic precipitator and not a “micronized” device. (This refers to particle milling). Also “componentry” is used to refer the vehicles and machines, not virus particles. Another word should be sought.
Authors concur with the reviewer. The title and references within the manuscript have been changed from “micronized” to “miniaturized”.
I have also noted that the efficiency for particle removal is slightly lower than the one for RNA as measured by RT-PCR, although very close. Is there any explanation for that?
The differences between the particle removal estimates and the viral RNA removal estimates can be accounted for in the method by which one measures each. Particle counts are measured by an analytical method that uses light scattering spectrophotonic technology (optical particle counter) whereas viral RNA was measured by RT-PCR which relies upon transformation of RNA that is converted to DNA and then amplified by annealing cycles after target primers are used. The expectation is that there will be slight differences in essentially the same measurement.

Round 2
Reviewer 1 Report
The manuscript has been improved, please consider the following comments:
- Please add a time series figure and particle size distribution data with error bars to show the stability of the system.
- What is the resolution time of the APS, was it set to 1 scan per second or else?
Author Response
Response to Reviewer (minor comments restated in italics for clarity)
Comment: Please add a time series figure and particle size distribution data with error bars to show the stability of the system.
A time-series figure displaying particle counts in the system is found in added suplementary data (S1) and particle size distribution data in the system is found in added supplementary data (S2). Both added figures are now referenced in section 2.6.2 Experimental Procedure.
What is the resolution time of the APS, was it set to 1 scan per second or else?
The resolution time (1 scan/second) has been added to resolution information, lines 224-227 of the manuscript.